# Reimagining Pharmacy Education through the Lens of a Choose Your Own Adventure Activity—A Qualitative Evaluation

**DOI:** 10.3390/pharmacy9030151

**Published:** 2021-09-06

**Authors:** Devin Scott, Alina Cernasev, Tyler Marie Kiles

**Affiliations:** 1Teaching and Learning Center, College of Pharmacy, The University of Tennessee Health Science Center, Memphis, TN 38163, USA; dscott50@uthsc.edu; 2Department of Clinical Pharmacy and Translational Science, College of Pharmacy, The University of Tennessee Health Science Center, 301 S Perimeter Park Dr., Suite 220, Nashville, TN 37211, USA; 3Department of Clinical Pharmacy and Translational Science, College of Pharmacy, The University of Tennessee Health Science Center, Memphis, TN 38163, USA; tkiles@uthsc.edu

**Keywords:** student pharmacist, Choose Your Own Adventure (CYOA), realistic scenarios, pharmacy curriculum

## Abstract

Background: Successful pharmacy curricula expose students to a variety of teaching and assessment methods to prepare students for clinical practice. However, development of clinical decision-making skills is often challenging for learners. To meet this need, the Choose Your Own Adventure (CYOA) Patient Case Format was developed to enhance traditional paper patient cases by integrating problem-based and case-based learning to improve pharmacy student learning. The objectives of this evaluation were to qualitatively evaluate the CYOA case format. The qualitative assessment of the student pharmacist’s learning experience utilizing this novel patient case format was used to formulate a template for extrapolation to other disease states. Methods: Focus groups were conducted with second year Pharm.D. students enrolled at the University of Tennessee Health Science Center (UTHSC) College of Pharmacy. The focus groups were conducted in Fall 2020, beginning the week after they were exposed to the CYOA case format. The corpus of data was analyzed thematically to identify themes using inductive coding. To establish the validity of this evaluation, the team met to assess the consistency of the data reduction methods and guard against methodological issues that could influence and affect coding decisions. Results: Participants were recruited until thematic saturation was achieved. Out of 25 participants, 23 participants provided demographic information, with 74% identifying as female. Thematic analysis identified three themes: (1) *“It was just fun!”* (2) Empowering Pharmacy Students through Groupwork: *“Collaboration [is] going to be vital”* and (3) Meeting the Need for Real-Life Scenarios: *“This is a real person.”* Conclusions: The data highlight that there are numerous advantages of adopting the CYOA format for delivering applied pharmacotherapy content. The CYOA format presents students with a realistic scenario that is fun and engaging and challenges students to justify their decisions regarding patient care in a structured group environment.

## 1. Background

Exposing students to a variety of teaching and assessment methods allows them to communicate effectively, solve difficult cases, and contribute to an interprofessional environment. Student engagement in the learning experience positively influences learning. Studies have shown that “flipped” classrooms have helped students integrate and apply their knowledge to problem solving and higher-order thinking [1]. Other strategies to increase engagement, such as “serious gaming”, have also shown promise in improving student engagement in the classroom [2,3]. A recent meta-analysis of studies comparing digital game-based learning found that digital games can also significantly enhance learning [4]. Development of a gaming instrument to enhance the learning experience for pharmacy students may be valuable to stimulate critical thinking.

Problem-based learning (PBL) and case-based learning (CBL) are well-studied approaches to improving learning outcomes through small-group teaching. PBL tasks students with working in small groups to tackle problems, create solutions independently, and debrief with a larger group of peers. The open inquiry necessitated by the PBL teaching method is advantageous for students because PBL requires students to struggle with problems and explore solutions, which can enhance problem-solving skills [5]. That being said, medical students are reported to prefer CBL because of curriculum density [6]. The CBL approach is concerned with the development of clinical expertise and calls for educators to prepare detailed cases with correct answers in advance, while necessitating that an “expert” guides students through the case, eventually arriving at the predetermined “correct” answer. CBL is generally considered to be “teacher centered” with the educator leading discussion and guiding students towards the pre-determined “correct” answer, whereas the PBL approach is more “student-centered”, with students engaging with problems independent of the educator and devising and justifying their own solutions [6].

The Choose Your Own Adventure (CYOA) Patient Case Format combines CBL and PBL in an interactive online format. In the CYOA activity, small groups of students work through problems independently, arrive at clinical decisions as a group, and justify their decisions, while simultaneously being guided to the most “correct” answers by built-in survey logic and expert facilitation. In the Interprofessional Education and Clinical Simulation (IPECS) III Course at the University of Tennessee Health Science Center (UTHSC) College of Pharmacy, a traditional paper patient case regarding outpatient diabetes management was transformed into an interactive format utilizing QuestionPro (Dallas, TX, USA), a free online survey platform. This required course is designed to teach students a consistent approach to the pharmacist–patient care process and assess skills-based activities through the use of active learning, team-based learning and clinical simulation. In this course, pharmacy students practice problem-solving skills to develop patient-centered care plans to improve patient outcomes. Class activities demonstrate the integral role of the pharmacist as a member of the outpatient healthcare team. For this activity, second-year PharmD students were tasked with “solving” progressively detailed and in-depth portions of an unfolding fictional, realistic patient case. The objectives of this activity were to improve knowledge of therapeutic management of type II diabetes and to enhance clinical decision-making skills through experience in outpatient diabetes management of a complex patient. Students worked in virtual small groups (using Zoom breakout rooms) and were facilitated by a senior teaching assistant. As students progressed through the evolving case, they were asked to select from a menu of plausible clinical decisions and come to a consensus. Once a clinical decision was reached and students input their rationale into QuestionPro, they were routed to a page that described the outcome of their choice. Each outcome provided an explanation as to why or why not that decision was the most correct choice. Student groups were encouraged to explore the outcomes from different clinical decisions, especially if their initial choice was not ideal. There was only one series of choices that eventually led to the most positive patient outcome. Along with text, the CYOA activity also incorporated images and gifs along each pathway to promote engagement. The groups were intermittently called back to the main Zoom room at designated points along the activity to discuss and justify their clinical decisions with the course instructor. While attendance in the IPECS III course is mandatory, students did not receive a grade for participation in this activity. This activity was delivered over one 50-min class period and was not made available to be repeated individually.

The purpose of this project was to qualitatively evaluate the CYOA case format. Participants were asked if they would recommend continuing to conduct the CYOA activity. Furthermore, participants were tasked with supporting their judgement through an evaluation of the CYOA activity in terms of student experience, teamwork aspects of the activity, and perceived impact on their decision-making skills. The qualitative assessment of the pharmacy students’ learning experience utilizing this novel patient case format was used to formulate a template for extrapolation to other disease states.

## 2. Methods

### 2.1. Study Design

The focus group methodology is widely employed by companies who want to test a new product or concept and facilitates feedback from different participants [7]. This evaluation used focus group methodology to facilitate group-thinking and brainstorming of student pharmacists’ opinions regarding their perceptions of the CYOA activity, commenting on each other’s experiences and points of view, and thus, generating new ideas [7]. This is an advantage because it provides a broader range of information [7,8]. The semi-structured focus group guide was designed to collect student perceptions of the advantages and disadvantages of various aspects of the activity and to provide recommendations for enhancement. Thus, the focus group guide was tailored to assess pharmacy students’ specific views regarding participation in the novel small group virtual activity.

### 2.2. Recruitment and Data Collection

The evaluation was approved by the University of Tennessee Health Science Center (UTHSC) Institutional Review Board. An email was forwarded to all P2 students (Nashville, Knoxville, and Memphis campus) explaining the evaluation purpose and inviting them to participate voluntarily in a focus group discussion. The interested participants contacted an external faculty specialist in education (DS) who conducted all the focus groups. To avoid any perceived biases, status, and influence relative to the participants, this evaluation used an external faculty who specializes in education to facilitate all the FG [9]. The Principal Investigator (PI) was a male instructional consultant at the UTHSC Teaching and Learning Center (TLC) who had no prior relationship with the participants and had no role in delivering the course. The focus groups were conducted one week after the students took part in the CYOA activity. Participants received a USD 20 gift card as compensation for their time. Due to the COVID-19 pandemic, all the focus groups were conducted via Zoom [8,10].

At the beginning of each focus group, the PI introduced himself to the participants, speaking of his role as an instructional consultant at the TLC, and informed the participants of the reasons for conducting this research. Then, the PI read the informed consent form and all the participants received information about their rights in this project [7]. All the participants agreed to participate in the evaluation and allowed the PI to audio record all the focus group discussions. The participants agreed to maintain confidentiality of their peers and not to discuss individual responses in session outcomes with anyone [7]. Furthermore, no other research team members were aware of the identities of focus group participants. The audio recorded focus groups were transcribed verbatim by a professional transcriptionist to avoid any biases. During the focus group discussions, the PI asked for clarifications when responses were unclear to ensure the nuance of the situation was correctly understood. The PI took field notes during data collection to note non-verbal expressions and interactions that were used in writing memos in the data analysis process [11].

A semi-structured focus group questions guide was used to facilitate discussions about student experiences with the CYOA activity [7]. This strategy allowed the PI to pose largely the same questions; however, the later focus groups incorporated additional questions, which were raised by earlier discussions [12]. In later focus groups, participants were asked more clarifying questions, including questions pertaining to suggestions for improvement of the CYOA activity, questions about what made group work un/rewarding, and questions about the real-world applicability of the CYOA diabetes activity. The scope of qualitative research and thematic analysis allows in-depth information to be gathered from the participants on different issues [12]. Furthermore, the strategy to incorporate additional questions leads directly to an enhancement of the external validity of the evaluation findings [9]. The relevant questions for the aim of this study are provided in Appendix A.

### 2.3. Data Analysis and Rigor

Thematic Analysis as described by Clarke and Braun was used to identify, analyze, extract, and report the themes from this evaluation [13]. Two researchers (DS and AC) read through the deidentified transcripts independently to familiarize themselves with the corpus of data. After familiarization with the data, the same researchers used an inductive approach to analyze the data [13]. The researchers coded inductively line-by-line all transcripts to identify, arrange, and systematize the emerging codes and categories [14]. During the inductive coding process, each researcher wrote memos that noted and facilitated distinct concepts in their initial code categories [11,15]. To determine consensus of the inductive codes and thematic analysis, the researchers team met together weekly. A third investigator (TK) joined the team to arbitrate differences between codes and refine themes per recommended technique [16]. For example, the team performed credibility checks to ensure there were not discrepancies in the identified codes [16]. To establish the validity of this evaluation, the team met to assess the consistency of the data reduction methods and guard against methodological issues that could influence and affect coding decisions [17].

The researcher’s meetings served to uncover nuances of the corpus of data and extract the less obvious content by asking themselves “What thoughts are the students trying to convey?” These reflections of the corpus of data were captured by writing memos that allowed the researchers to reflect on their own thoughts to construct an understanding of the participant environment. [16] Furthermore, the reflexivity was not only vital for capturing the meaning of the data, but also the scope of this evaluation was to focus on the utility and possible extension of the CYOA activity and its application to other courses in the College of Pharmacy [18].

The Thematic Analysis process ensured that the emerging themes were comprehensive, capturing all key elements from the corpus of data, relevant and conceptually congruent to the scope of this evaluation. For accuracy, the team was in consensus and agreed on final theme labels. The team followed the recommendations from Braun and Clarke, and Lincoln and Guba, regarding the point of saturation when the team agreed that there was no more additional information to extract [19,20].

## 3. Results

The CYOA activity was administered to P2 students enrolled in the IPECS III course. Four focus groups were conducted with 25 participants in September 2020. Out of 25 participants, 23 participants provided demographic information, with 65% indicating that they grew up an in urban environment and 74% identifying as female. Participants self-described their race as the following: 48% White, 35% African American, 13% Asian, 4% Other.

Three themes emerged from analysis that relate to the concurrent CBL and PBL elements of the CYOA case design. The first theme explores the concept of fun and engaging learning. Participants frequently asserted that they paid greater attention to course content because the CYOA activity was fun and engaging. The second theme relates to group dynamics discussed by participants in relation to the CYOA activity. Participants emphasized the benefits of participating in groups in relation to preparation for membership in healthcare teams and learning from a diversity of perspectives. The final theme encompasses the applied aspects of the case design, with participants referring to real-life situations. Participants stressed that the CYOA activity was more realistic than traditional cases and that the enhanced realism better prepared them for “real world” pharmacy practice than lectures or traditional cases.

### 3.1. Theme 1: “It Was Just Fun!”

This theme described the participants perspective that the CYOA activity that was enjoyable as well as educational. Participants’ focus on the fun aspects of the activity highlighted the increased attention students lend to engaging in learning activities. Furthermore, participants indicated that engaging activities lead to a deeper understanding of the course material.

Participant 4 (P4) continuously emphasized the theme of fun and engaging: 

“[The CYOA activity] was just fun. And it was funny. Like I was laughing… I can tell that like they put effort into it. It was something that just as much as it was important for the information to get through to us, that we had fun doing it.” (FG1, P4) 

P4 asserts that fun and funny teaching activities demonstrate the instructors’ commitment to helping students meet learning objectives, which, in turn, motivates students to attend to content more closely. P4 then references the limited attention span of learners who have been in lectures for hours:

“If it’s been a whole morning of lectures, I’m exhausted…my attention span may not all the way be there. I may be a good two cups of coffee into it, and so I’m just ready to just like shut down. And so, in that sense, having something there where there’s like a funny scenario or a funny picture or just like- just a funny scenario in general just makes my heart a little bit lighter and my attention span just a little bit longer. So, I can appreciate the effort that they put into that.” (FG1, P4)

In this quotation, P4 again highlights the benefits of engaging and fun activities, stating that activities such as CYOA have the potential to re-energize learners and refocus them on class content. Next, P4 contrasts the CYOA activity with standard didactic lectures:

“Standard lecture is like literally just giving your child a plate of broccoli and be like, eat it, you got to eat it, it’s good for you… We’ll get through it, and once it’s over, hallelujah. Versus the activity with that are kind of like those broccoli tater tots. It’s the same stuff, but the way that it’s presented and it’s in a fun, cool way that’s more appetizing, then that’s something that I would prefer.” (FG1, P4)

P4 indicates that the method of content delivery matters and that engaging activities are preferable to standard lectures. Finally, P4 critiques standard lectures and declares that fun activities such as CYOA are not only preferable to lectures, but that they are perceived to be more relevant to the learning needs of Pharmacy students:

“It was a fun way, it made good use of my time, which when we’re trying to get 40 hours’ worth of material, [I’m] a stickler about my time, and I feel a lot of lecturers or standard base way of teaching is a- it’s a waste of our time and a waste of energy. So, in [the CYOA] activity, I appreciate that it was fun… it was a good use of my time.” (FG1, P4)

P5 echoes the sentiments of P4:

“[normally] it’s just a bunch of cases and we go through them every Friday… So, …having something like this where it is a bunch of information and we listen to over ten hours of lecture each week, it’s nice to switch it up and be like, hey, let’s have something fun and active to apply this information, so that we can apply it but also remember it.” (FG1, P5)

Again, P5 argues that the fun and engaging aspects of the CYOA activity are not only preferable to standard lectures but that they also lead to deeper engagement with course content.

P7 agrees and recommends that the CYOA format would be beneficial for teaching a variety of classroom material:

“It’s really good help to learning and it would be something that’s easy to apply to any case, hypertension, heart failure- I mean, you could throw it in any scenario and this activity would work.” (FG2, P7)

Finally, P2 argues that more Pharmacy educators should adopt the CYOA activity format, because it is enjoyable and leads to greater engagement with course content:

“I loved the activity. It was great, and I think more teachers should incorporate things like that because it’s not as boring as just sitting there looking at a patient case where it’s literally black and white, A, B, C, D, no explanation really… this is what I was looking for. Well, why? I need more information.” (FG1, P2)

Thus, participants indicated that the CYOA activity was fun and enjoyable. These aspects of the activity were perceived to lead to increased student engagement and enhanced learning. Participants who found the activity to be fun and enjoyable were likely to recommend the CYOA activity format be implemented throughout the pharmacy curriculum. Ultimately, participants highlighted that the CYOA format led to deeper learning because students were actively engaged in learning as opposed to passively receiving information from lecturers or participating in unoriginal case-based learning.

### 3.2. Theme 2: Empowering Pharmacy Students through Groupwork: “Collaboration [Is] Going to Be Vital”

This theme highlights participants’ appreciation for the opportunity to discuss aspects of the CYOA activity case in groups, in part, due to the opportunity to consider multiple perspectives and discuss with their peers. Furthermore, students indicated that the structured groupwork featured in the CYOA format led them to consider their future role as members of the healthcare team.

P8 felt as though the group aspect of the CYOA activity was beneficial to their learning:

“I thought that the teamwork aspect was one of the high points of this activity. Because in previous IPECS sessions, when we have the actors come in, it’s kind of one on one with the rest of the group kind of staring, and it kind of- I don’t know, it isolated me in my own mind. But this was a lot of teamwork. And it was an enjoyable experience.” (FG2, P8)

P9 echoed the same message:

“I agree with what [P8] is saying. Having a team of people go over the entire case with you helps you get into a different mindset and helps you see what other people are thinking as well.” (FG2, P9)

P10 also indicated that they appreciated the group aspect of the CYOA activity:

“I think it’s just positive in general because not everyone is going to have the same opinion, especially on a problem that can be answered in multiple different ways, so it’s nice to hear other opinions and kind of group that together for a common concept.” (FG3, P410)

P13 also highlighted the structured groupwork, comparing CYOA groupwork to traditional group CBL. P13 says:

“I would just say I thought it was successful… I think, with this activity, everybody was able to read at the same time, talk about what they thought, do you agree or disagree, and then move on. And so, I felt like I could actually go along at the same pace as everybody on this activity instead of feeling like everybody was at all different points of an assignment and like trying to keep up… I felt like it was successful.” (FG3, P13)

Furthermore, P21 described the collaborative aspects of the groupwork and emphasized the realistic nature of the CYOA activity:

“I like that it allowed us to explore the gray parts of pharmacy, like was said earlier. And also, we were either encouraged to either make mistakes or go back and explore why certain answers were wrong, just to kind of get that understanding behind it. And then, also, it was really hands-on and collaborative with our groups.” (FG4, P21)

Here, P21 indicates that the opportunity to discuss a realistic case with group members, where students were encouraged to explore different “answers” led to more engaged and deeper learning. 

In the quotation below, P13 mentions that the groupwork and realistic scenario called attention to the relationships pharmacists develop with patients and colleagues in practice:

“I think the activity was successful in, [sort of] narrowing in on that aspect of interpersonal relationships with your colleagues and [the] patient.” (FG3, P13)

P20 echoes P13′s sentiments and argues that pharmacy students should engage in learning experiences that enhance their ability to communicate with colleagues, healthcare teams, and patients: 

“I really enjoyed the teamwork aspect because, as pharmacists, we need to go ahead and work towards being able to communicate with nurses, communicate with our patients, communicate with physicians. So going ahead and having that practice with communicating with other pharmacists and working on that collaboration is something that I think is going to be vital for when we actually get out into practice.” (FG4, P20)

The structured groupwork facilitated in the CYOA activity was viewed positively by focus group participants. Participants perceived improved learning outcomes related to course content that they attributed to the open exchange of ideas among group members. Furthermore, participants indicated that the group discussions and focus on realism in the CYOA activity afforded them the opportunity to reflect on the communication skills that will be expected of them upon matriculation as they relate to patients, colleagues, and the healthcare team. The CYOA activity was perceived to reinforce effective communication strategies for engaging with a healthcare team while simultaneously providing for a variety of perspectives that enhance learning.

### 3.3. Theme 3: Meeting the Need for Real-Life Scenarios: “This Is a Real Person.”

This theme describes the real-life applicability of what students learned during the CYOA activity. Participants presented the relatability of the diabetes case discussed in the CYOA activity in contrast to the traditional cases they encounter elsewhere in their education. Furthermore, participants valued the opportunity to engage with the “gray” areas of applied pharmacotherapy where they considered the socioeconomic status and history of the patient when recommending treatment options.

P12 indicated that the rich background description made the CYOA case activity relatable: 

“I liked how, with patient cases we knew… their money status, their life, their home status, and I think that can be really relatable to real life… out working in the community setting. So, I thought that was very beneficial.” (FG3, P12)

P12 felt as though the additional background knowledge made the case relevant to them and their future practicing pharmacy in the community.

P11 adds:

“[The CYOA activity] did have certain pathways where the patient would indicate, oh, well, it’s my son’s birthday, I want to have cake. And we’re not going to tell somebody, oh, you can’t have cake because you have diabetes. So little things like that did sort of remind us that, oh, this is a real person. They’re not going to restrict their diet the way we tell them to every single day for the rest of their life just because of this disease.” (FG3, P11)

P11 indicates that background information about patients led them to a richer understanding of the complexity of treating disease states such as diabetes. P11 was reminded that upon graduation, they would be working with patients who will not always adhere to prescribed regimes. P2 elaborates further on relating to patients, also referencing the cake example as a marker of the perceived applicability of the CYOA activity to real-life:

“I think the best part is the fact that it’s a real-life scenario. It’s not textbook, this is your specific answer, and you go in knowing that not everyone’s answer might be the same, but you still have an idea of, okay, if you actually were presented with this patient as a pharmacist, you’re still able to look at your options for how you could treat them… let’s be realistic. If somebody wants to eat cake, let them eat cake, but you can educate them on how to go about eating cake, like eating something smaller or having to up their insulin dose… some people don’t think about that, and I think the realness of a case brings that awareness to people who don’t necessarily know much about people who do have diabetes or how to treat them.” (FG1, P2)

P2 felt as though the CYOA activity presented realistic cases that made them think about the lived experiences of patients with diabetes. Furthermore, P2 appreciated the opportunity to reflect on different strategies for working with patients with diabetes.

P13 further explores the real-life theme:

“It’s not black and white in the real world. And I know that they tell us that, but, I mean, when we’re in school, our answers have to be, in a sense, black and white for tests and exams, and so I think this was just a nice way to… explore your options and understand why you could go multiple routes and reason through it and figure out what you think would be best. And so, I just thought that this activity did a really good job of that.” (FG3, P13)

Here, P13 differentiates the CYOA activity from tests and exams, speaking to the exploration of different treatment options that more accurately reflects the real-world outside of pharmacy school. 

P14 also felt as though the CYOA format more accurately reflected the real world than traditional examinations:

“I feel like this was… like how I would think naturally in the real-world setting. And, at the end of the day, this is a real patient case, but for the exam, it’s still an exam question. There’s not a patient, it’s just an exam question. But for this one, it’s like a real person that has real medications and a real issue, so that was definitely a real-world experience that I liked.” (FG3, P14)

Relatedly, P11 said:

“I think it definitely helped us to take a step back for a moment on just strictly thinking I want the best grade, so I’m going to pick the answer based on what I think the professor will like, and it allowed us to take more accountability for what we personally think…not only that but also tie it into like that patient-health provider relationship. Because even if I do think something is correct, that patient may not want it. And so, we always have to accept what they’re actually going to go with. Because, at the end of the day, these are just our recommendations, and it’s not something they have to follow. So, it really did give us that opportunity to just see them as a person and try to put that first before all else.” (FG3, P11)

P11 differentiates the CYOA activity from other classroom assignments and examinations, emphasizing that the activity allowed students to take accountability for how their recommendations would be received in the real world. P8 valued the opportunity the CYOA activity gave students to think about the responsibilities pharmacists hold in the patient–healthcare provider relationship in relation to a realistic case. Thus, participants stated that the CYOA activity more accurately represented the real world than traditional cases and examinations. Furthermore, many participants felt as though the real-world aspects of the CYOA activity enabled them to reflect on the experiences of people with diabetes and think about their role as healthcare providers in the community. The realistic aspects of the scenario featured in the CYOA activity led to deep engagement with course content, facilitated in-depth discussions among pharmacy students, and challenged students to consider the rationale behind the “right” and “wrong” answers.

## 4. Discussion

The CYOA activity is a novel approach to pharmacy instruction that pairs case-based learning with problem-based learning, asking students to work in groups to work through a realistic scenario. Similarly, to PBL and CBL activities, the CYOA format enhanced student engagement and was well-received by students. [5,6] Thus, the CYOA activity is a unique approach to delivering content that can be leveraged throughout pharmacy curricula. Support for broader adoption of the CYOA format is supported by three themes that emerged from discussions with student pharmacists. 

Focus group interviews were conducted to decide if the CYOA format should be continued in its current form. Of particular interest to the research team were the rationales given for final judgement. When asked to evaluate the CYOA format and render judgement, participants recommended that the CYOA format should be adopted throughout the pharmacy curriculum. 

In terms of the student experience, not only did participants indicate that this format is more enjoyable than traditional cases or lectures, but they also indicated that they engaged more deeply with course content because they were having fun. Contemporary research on purposeful gamification supports this notion, with fun learning experiences leading to increased learner motivation and engagement, and, ultimately, better learning outcomes. [21] Engaging students through novel and gamified learning experiences can lead to enhanced learning and engagement with pharmacy content.

The teamwork aspects of the CYOA activity were well received. Participants valued the structured, unfolding CYOA scenario, which allowed them to actively engage in discussion with their group members. They appreciated the diversity of perspectives and the opportunity to practice and reflect on the communication skills that will be required of them as they matriculate and join healthcare teams. The development of empathetic communication skills among healthcare professionals, wherein healthcare professionals comprehend the personal experiences of patients, is recognized as an important skill for improving experiences and outcomes for patients and providers. [22] By providing opportunities for purposeful, directed groupwork, the CYOA activity offered pharmacy students the opportunity to direct their learning, engage in thoughtful discussions about course content with their peers, and reflect on their role on the healthcare team.

According to participants, the realistic aspects of the CYOA activity affected their decision making by challenging them to consider the real-world implications of their decisions. The realism of the CYOA activity identified by participants further enhanced learning by offering a scenario that encouraged students to consider the life experiences of patients. The realistic aspects of the CYOA activity had students agonizing over whether or not to forbid a patient with diabetes from enjoying a slice of cake at a birthday celebration. Furthermore, the detailed background and unfolding scenario challenged students to convince their peers of the correct “answers” to provide the best patient care. Students were tasked with providing justifications for their decisions and were likewise given rationales behind “correct” and “incorrect” answers. Participants stressed that the realistic aspects of the CYOA activity caused them to reflect on their role in the community as healthcare professionals in addition to engaging purposefully with the content offered in the scenario. This finding aligns with research on virtual patients, which were found to elicit emotional responses among students and to appropriately introduce students to the complexities of real-life healthcare. [23] Ultimately, the realism offered by the CYOA activity facilitated the fun and engaging aspects of the activity, while providing the depth necessary for worthwhile group discussions. 

The CYOA activity can be adapted and applied to a variety of disease states. Figure 1 provides a template for developing CYOA activities based on the layout and flow of the diabetes management case discussed here. The sections outlined in bold were the designated points for large group discussion. Using this template, educators may convert existing cases to CYOA activities to engage students in fun, reflective, group learning.

The CYOA format is an engaging new approach for pharmacy educators who wish to involve students in realistic scenarios that lead to meaningful engagement with course content and reflection upon the pharmacy professional’s role in the community and on the healthcare team.

## 5. Strengths and Limitations

To our knowledge, this is the first evaluation to use a qualitative approach to evaluate the CYOA case format. This qualitative evaluation provides rich and contextualized experiences of second-year student pharmacists’ exposure to the CYOA format that have resulted in the development of a template activity for Pharmacotherapy courses and future research. Furthermore, consideration was made throughout the evaluation to enhance the validity and trustworthiness of the findings. For example, an external facilitator conducted all the focus groups to avoid perceived biases, status, and influence relative to the participants. However, the sample predominantly comprised pharmacy students from only one College of Pharmacy. Future studies might consider including pharmacy students from other colleges or other healthcare students that use CYOA for their didactic courses in order to refine the CYOA format.

## 6. Future Research

Future research related to the CYOA format may include exploration of modifications related to group size/makeup, individual completion, as well as presence or absence of a group facilitator. Future investigation of the CYOA format should also include multi-institution studies, examination of CYOA’s addition to other applied pharmacotherapy courses, as well as the role of CYOA in developing a robust curriculum. While assessment of the efficacy of this activity may be difficult to achieve through examination, appropriate endpoints for future projects may include student performance on written patient cases or during experiential rotations. Another avenue for research would consider CYOA’s role in preparing pharmacy students to matriculate into the role of members of the healthcare team.

## 7. Implications

This evaluation explored pharmacy students’ experiences with a Choose Your Own Adventure activity format. The CYOA format combines an unfolding, realistic scenario with structured groupwork to engage students in learning, while requiring students to articulate rationales for their decisions. From this investigation, the CYOA format can be recommended for implementation in applied pharmacotherapy courses where faculty privilege deep learning, structured groupwork, and reflection on the role of pharmacy practitioners on the healthcare team. 

## 8. Conclusion

This qualitative evaluationdemonstrated the benefits of adopting the CYOA format for delivering applied pharmacotherapy content. The CYOA format presents students with a scenario that is fun and engaging and challenges students to justify their decisions regarding patient care in a structured group environment. Participants emphasized that they engaged deeply with the course content and were encouraged to view patients holistically and reflect on their roles and responsibilities as healthcare providers upon matriculation. We believe that the CYOA format should be considered when delivering content in applied pharmacotherapy courses and when preparing a robust curriculum. The CYOA format can be leveraged to improve the student learning experience and to encourage students to engage deeply with course material.

## Figures and Tables

**Figure 1 pharmacy-09-00151-f001:**
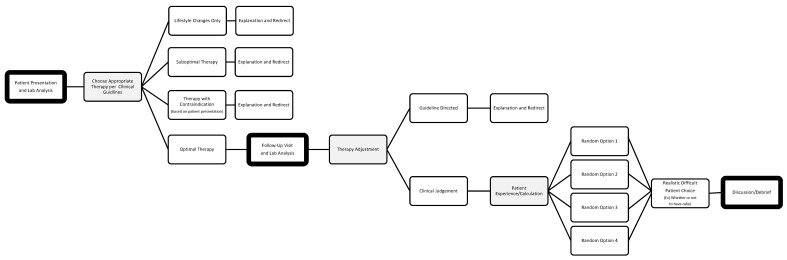
CYOA activity template for pharmacotherapy courses.

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
