# Peer review of "Reimagining Pharmacy Education through the Lens of a Choose Your Own Adventure Activity—A Qualitative Evaluation"

_pharmacy, 2021, doi:10.3390/pharmacy9030151_

Round 1
Reviewer 1 Report
Dear Authors,
Thank you for the opportunity to review your paper, “Reimagining Pharmacy Education Through the Lens of a Choose Your Own Adventure Activity: Student Pharmacist Perceptions - A Qualitative study”. The overarching objectives of this work were “to test and develop the CYOA case format”. This paper introduces an innovative approach to teaching and learning and will be of interest to pharmacy educators. My sense was that the overall aim of this work may be to evaluate the CYOA approach using a qualitative method to gather data versus a research study to understand students’ learning experiences. This distinction can be clarified by the overall goal and objectives for this work. Depending on how you decided to frame the paper will guide your decisions in the revision process. For a qualitative research, I suggest using the COREQ criteria to guide reporting (see Tong A, Sainsbury P, Craig J. Consolidated criteria for reporting qualitative research (COREQ): A 32-item checklist for interviews and focus groups. Int J Qual Health Care. 2007;19:349–357.) I offer the following suggestions to enhance the presentation of your work. Some of my comments are through the lens of qualitative research and may or may not be applicable depending on how the paper is positioned (evaluation of an educational technique versus study of students' experience).
Abstract
Clearly state the research question (objective) for this qualitative research study. Clarify if the overall purpose was to evaluate the COYA case format, or to explore students’ experiences. (Check to ensure the objective is stated consistently throughout the paper, see Lines 17, 87, 93, 148, 164, 438)
Ensure the conclusions address the research question (objective). As written, the conclusion addresses the implementation of CYOA. If the research question relates to student experiences, consider revising.
Introduction
Include more details of the context / background for the study. This could be an additional paragraph describing the pharmacy program, the course, and the case. Describe the students, e.g., class size, how groups were formed, group size, etc. Providing these details will assist reader in duplicating the approach and establishing transferability to their own program or course. Consider including details of the CYOA case, along with instructions for students and facilitation guides, in an appendix.
Line 72: Suggest including a reference for Question Pro.
Methods
Line 91: Provide rationale for using focus groups to collect data. Include the focus group guide and updates.
Line 110: Clarify the PI’s relationship with the students and the course. Describe the research team (see COREQ).
Line 129: Consider including a table of codes, categories, and themes.
Results
Line 158: Consider including a table to highlight details of each focus group (e.g., number of participants, dates).
Overall, the results are presented in detail highlighting individual’s contributions. I wondered if the analysis captured discussion of participants within each focus group (e.g., it is not possible to know if P4 and P12 were in the same group) and if there were any differences between groups.
Discussion
I encourage expanding the discussion section to link back to the research question (objective) and the issues identified in the introduction (e.g. PBL, CBL, CYOA). Incorporate literature on students’ experiences with these types of activities.
Address trustworthiness of this qualitative research (e.g., see Amin MEK, Norgaard LS, Cavaco AM, et al. Establishing trustworthiness and authenticity in qualitative pharmacy research. Res Social Admin Pharm. 2020;16:1472-1482.). Include the strengths and limitations of this work.
Conclusion
Consider reframing to focus on the research question (objective) of the study and the method.
Author Response
Abstract
Clearly state the research question (objective) for this qualitative research study. Clarify if the overall purpose was to evaluate the COYA case format, or to explore students’ experiences. (Check to ensure the objective is stated consistently throughout the paper, see Lines 17, 87, 93, 148, 164, 438)
Response: Thank you for this point of clarification, this objective of the paper (to qualitatively evaluate the CYOA case format) has been clearly stated in the abstract and reinforced throughout.
Ensure the conclusions address the research question (objective). As written, the conclusion addresses the implementation of CYOA. If the research question relates to student experiences, consider revising.
Response: Thank you for this clarification. The conclusions as written reflect the clarified objective of the paper.
Introduction
Include more details of the context / background for the study. This could be an additional paragraph describing the pharmacy program, the course, and the case. Describe the students, e.g., class size, how groups were formed, group size, etc. Providing these details will assist reader in duplicating the approach and establishing transferability to their own program or course. Consider including details of the CYOA case, along with instructions for students and facilitation guides, in an appendix.
Response: We value your suggestion. As this work posits, the CYOA patient case format may be extrapolated to a wide range of courses/content. The authors believe that the details from the student experiences are not limited to the type of program, course, case or class/group makeup, and therefore have not provided these details. The template provided in Figure 1 should allow this activity to be adapted to any program, however, this consideration has been incorporated in the recommendations for future research section (line 442)
Line 72: Suggest including a reference for Question Pro.
Response: Thank you for this suggestion. QuestionPro location has been cited in this revision (pg 5 Line 101)
Methods
Line 91: Provide rationale for using focus groups to collect data. Include the focus group guide and updates.
Response: We value your suggestion. The text has been amended.
Line 110: Clarify the PI’s relationship with the students and the course. Describe the research team (see COREQ).
Response: Thank you for asking for further clarification about the research team and the PI’s relationship with students. Clarifications have been added on lines 137, 140, & 474.
Line 129: Consider including a table of codes, categories, and themes. Response: Thank you for this suggestion. The research team choose not to include a table of the codes, categories, and themes since there is an opportunity to publish another manuscript with the data.
Results
Line 158: Consider including a table to highlight details of each focus group (e.g., number of participants, dates).
Response: We value your suggestions. The authorship amended the text by providing additional information regarding the number of participants and dates.
Discussion
I encourage expanding the discussion section to link back to the research question (objective) and the issues identified in the introduction (e.g. PBL, CBL, CYOA). Incorporate literature on students’ experiences with these types of activities.
Response: Thank you for this recommendation. A statement has been added to the discussion to connect this format to the introduction (Line 395-396). As this is a unique combination of both types of activities, the CYOA format remains the focus of the discussion.
Address trustworthiness of this qualitative research (e.g., see Amin MEK, Norgaard LS, Cavaco AM, et al. Establishing trustworthiness and authenticity in qualitative pharmacy research. Res Social Admin Pharm. 2020;16:1472-1482.). Include the strengths and limitations of this work.
Response: We value your suggestion. We addressed trustworthiness and also included strengths and limitations of this work.
Conclusion
Consider reframing to focus on the research question (objective) of the study and the method.
Response: Thank you for this comment. The conclusions as written reflect the clarified objective of the paper.as noted above.
Reviewer 2 Report
The methods of this qualitative study are sound and would be of interest to other educators interested in performing focus groups.
Methods
- More description of the CYOA activity is warranted. How long was the activity? Was it a group activity? Could students repeat it? Was it graded? Was it required?
Results
- What questions were new in the later groups?
- I think the results should be simplified and student quotes removed. But this is a personal suggestion, not scientific.
- Were there only 3 themes? What were other less common themes?
Discussion
- Missing from the discussion section was how the CYOA activity should be modified based on data gathered from focus groups.
- Figure 1 is very helpful.
- Future directions should include assessment of efficacy of this activity as a learning methodology right?
Author Response
Methods
- More description of the CYOA activity is warranted. How long was the activity? Was it a group activity? Could students repeat it? Was it graded? Was it required?
Response: Thank you for this comment. The activity was conducted in small groups as described. Details regarding the length, requirements and grading have been included in this revision. (Pg 5 Line 114-116)
Results
- What questions were new in the later groups?
Response: We value this excellent suggestion. The types of questions added to later groups have been added to the manuscript, starting on line 161.
- I think the results should be simplified and student quotes removed. But this is a personal suggestion, not scientific.
Response: Thank you for this suggestion. Since the other reviewers did not recommend the removal of the quotes, we decided to keep the quotations. Furthermore, the strength of this manuscript stays in the quotations.
- Were there only 3 themes? What were other less common themes?
Response: Thank you for the clarifying question. We plan in disseminating another manuscript that has different aim. Thus, we cannot publish all the data in this manuscript. Thank you for your understanding.
Discussion
- Missing from the discussion section was how the CYOA activity should be modified based on data gathered from focus groups.
Response: Thank you for this suggestion. A statement has been added to this portion of the manuscript in this revision (Line 442)
- Figure 1 is very helpful. Thank you for this comment!
- Future directions should include assessment of efficacy of this activity as a learning methodology right?
Response: Thank you for this comment. A statement has been added to the future research section (pg 21, Line 441-443)
Reviewer 3 Report
No additional comments or suggestions. Clear and well written paper.
Author Response
No additional comments or suggestions. Clear and well written paper.
Response: We value and appreciate your time to review our manuscript.
Round 2
Reviewer 1 Report
Dear Authors,
Thank you for the opportunity to review your revised paper, “Reimagining Pharmacy Education Through the Lens of a Choose Your Own Adventure Activity - A Qualitative study”. The overarching objectives of this work were revised “qualitatively evaluate the CYOA case format”. I believe this work will be of interest to readers of Pharmacy. Building on my previous review, further revisions will strengthen the presentation of your work.
Issues that must be addressed:
- As the overall objective is to evaluate the case format, reframe this work as an evaluation (not a qualitative study).
- Purpose: Clarify the overall aim of the evaluation, e.g. to improve the format, or identify needed enhancements (formative)? to decide if the format should be continued (summative)?
- Objectives: Indicate specific objectives that were qualitatively evaluated (i.e., identify specific aspects of the format, skill development -- decision-making skills, communication, teamwork, student experience, etc.).
- Introduction: Describe the course, course objectives, case objectives. Clarify if the case was fictional or a real-life case scenario. Describe the students enrolled in the course (since it is and Interprofessional Education and Clinical Simulation course, were there students from other disciplines enrolled?).
- Method: Append the focus group interview guide.
- Discussion: State the numerous advantages of the CYOA. Emphasize what is unique about this approach and why it should be continued or adapted. Outline any changes planned for the CYOA case format , or future evaluations, arising from this evaluation. Discuss this unique case format developed for this course in relation to other CYOA approached in the literature.
Issues that should be addressed:
- Method: Append the case study
Author Response
Issues that must be addressed:
- As the overall objective is to evaluate the case format, reframe this work as an evaluation (not a qualitative study). Response: Thank you for this comment. The title has been edited to reflect the aims of the work. Additional wording to several sections has been added to slant more towards evaluation as well.
- Purpose: Clarify the overall aim of the evaluation, e.g. to improve the format, or identify needed enhancements (formative)? to decide if the format should be continued (summative)? Response:Thank you for this suggestion. The work focused on summative judgement with an interest in rationale for judgement. This is reflected in our data as well as the discussion section.
- Objectives: Indicate specific objectives that were qualitatively evaluated (i.e., identify specific aspects of the format, skill development -- decision-making skills, communication, teamwork, student experience, etc.). Response: We appreciate this feedback. The wording in this section has been modified to highlight the evaluation aspect and center these specific objectives in the discussion section.
- Introduction: Describe the course, course objectives, case objectives. Clarify if the case was fictional or a real-life case scenario. Describe the students enrolled in the course (since it is and Interprofessional Education and Clinical Simulation course, were there students from other disciplines enrolled?). Response: Thank you for this clairifcation. The introduction has been modified to reflect the course and activity objectives, to clarify that the students were pharmacy students only, and that the case was fictional and realistic.
- Method: Append the focus group interview guide. Response: Thank you for this feedback, the focus group guide has been provided as an appendix.
- Discussion: State the numerous advantages of the CYOA. Emphasize what is unique about this approach and why it should be continued or adapted. Outline any changes planned for the CYOA case format , or future evaluations, arising from this evaluation. Discuss this unique case format developed for this course in relation to other CYOA approached in the literature.
Response: Thank you for these suggestions. The authorship team reviewed them, and we would like to highlight that these advantages are already pretty prevalent in the discussion and implications section. Furthermore, the results section features comments where students compare CYOA to traditional cases.
Issues that should be addressed:
- Method: Append the case study: Thank you for this comment. This case study has not been included in the appendix due to intellectual property concerns.
Round 3
Reviewer 1 Report
Dear Authors,
Well done! Thank you for addressing the reviewer's comments. I noted that the title and objectives were revised to signal to the readers that that this paper reports a summative evaluation of a one-hour CYOA instructional patient case. The manuscript requires minor editing for format (e.g., font) and consider replacing 'study' with 'evaluation' throughout the manuscript for consistency.
Author Response
The manuscript requires minor editing for format (e.g., font) and consider replacing 'study' with 'evaluation' throughout the manuscript for consistency.
Response: Thank you for your time to review our manuscript. We made the changes in the manuscript. We replaced "study" with "evaluation" that strenghtened our paper. Thank you for this valuable suggestion.